# Factors Associated with the Severity of Pregnancy-Related Hypertensive Disorder: Significance of Clinical, Laboratory, and Histopathological Features

**DOI:** 10.3390/diagnostics12092188

**Published:** 2022-09-09

**Authors:** Hyo Jung An, Dae Hyun Song, Yu-Min Kim, Hyen Chul Jo, Jong Chul Baek, Hyoeun Kim, Jusuk Yang, Ji Eun Park

**Affiliations:** 1Department of Pathology, Gyeongsang National University Changwon Hospital, Changwon 51472, Korea; 2Department of Pathology, Gyeongsang National University School of Medicine, Jinju 52727, Korea; 3Institute of Health Science, Gyeongsang National University, Jinju 52727, Korea; 4Department of Obstetrics and Gynecology, Gyeongsang National University Changwon Hospital, Changwon 51472, Korea; 5Department of Obstetrics and Gynecology, Gyeongsang National University School of Medicine, Jinju 52727, Korea

**Keywords:** albumin, placenta, pregnancy-related hypertensive disorder, syncytial knots

## Abstract

The purpose of this paper is to evaluate the association of maternal clinical and laboratory features and placental histopathological changes with disease severity in pregnancy-related hypertensive disorders. From January 2021 to December 2021, clinical and laboratory data at the time of delivery and histopathological features of the placenta were collected from pregnant women with pregnancy-related hypertensive disorders at a single institution. The women were classified according to the pregnancy-related hypertensive disorder clinical severity, and each variable was compared accordingly. Gestational age-matched normotensive groups were also compared. Univariate and multivariate regression analyses were used to identify factors influencing pregnancy-related hypertensive disorder severity. Fifty-eight pregnancies were analyzed. Maternal albumin levels before delivery (beta coefficient −0.83, *p* = 0.043) and increased placental syncytial knots (beta coefficient 0.71, *p* = 0.026) are important parameters that are closely related to disease severity in women with pregnancy-related hypertensive disorders. The combination of albumin, PAPP-A, total bilirubin, and eGFR levels appears to be optimal for predicting pregnancy-related hypertensive disorder severity.

## 1. Introduction

Pregnancy-related hypertensive disorders, one example of which is preeclampsia (PE), is a leading cause of maternal morbidity and mortality worldwide [1,2], and women with PE have an increased long-term risk of premature cardiovascular disease and mortality [3]. In addition, PE increases the risk of negative perinatal outcomes for fetuses and newborns, which include intrauterine growth retardation, low birth weight, and stillbirth. Approximately 15% of all preterm births are due to birth indications for pregnancy-related hypertensive disorders [4]. PE, which is one of the representative pregnancy-related hypertensive disorders, is a multiorgan syndrome with new onset high blood pressure and proteinuria, which is related to placental pathology and results in hypertension and decreased multiorgan perfusion. Hemolysis, elevated liver enzymes, and low platelet count (HELLP) syndrome, the most severe form, have maternal mortality rates of 1–25% [5]. In addition, the prevalence of pregnancy-related hypertensive disorders is also increasing as the rates of older pregnant women and women with multiple pregnancies increase.

Currently, the only effective and fundamental cure for controlling maternal and fetal PE complications is optimally timed birth. Because the decision to terminate a pregnancy depends on the severity of the findings, previous studies have investigated biomarkers such as liver or kidney function, ultrasound results, or clinical parameters for the severity of the disease [6,7,8,9,10] and reported that these markers are affected by the onset and severity of PE. However, indications for immediate or delayed delivery depend largely on the clinician’s experience. Therefore, the more informative the markers for the criteria in judging PE severity, the more supportive the criteria can be in determining the time of delivery in clinical practice.

The association between PE and placental histopathological lesions, mostly represented by maternal vascular malperfusion (MVM) lesions, has been studied [11]. Only a few studies has investigated the association between the severity of PE and the findings of placental histopathology, even classifying it into two types, namely, mild and severe PE [12], or early-onset and late-onset PE [13], and none of these studies looked for a correlation with gestational hypertension without proteinuria and HELLP syndrome.

Although there is still a lack of consensus on the true pathophysiology of PE, experts support the hypothesis that PE is a primary placental disorder. The two-stage theory [14], which is a representative model for understanding the concept of PE, is explained by dividing PE into stage 1, a preclinical stage represented by poor placentation, and stage 2, a maternal clinical syndrome stage due to a hypoxic placenta. Therefore, in this study, we investigated whether there are significant differences in placental histopathology and clinical and laboratory variables through classification according to the severity of pregnancy-related hypertensive disorders, including normal pregnancy, and further identified meaningful markers related to severity.

## 2. Materials and Methods

### 2.1. Study Population

This study was performed at Gyeongsang National University Changwon Hospital from January 2021 to December 2021. All women included in this study gave birth at the research institution, and prenatal laboratory tests and postnatal placental histopathology were also performed at the same institution. The women’s ages ranged from 20 to 42 years and the weeks of gestation ranged from 26 to 39.6 weeks. Of the 58 pregnant patients included in the study, 44 patients with pregnancy-related hypertensive disorders were included in the study group, and 15 gestational age-matched patients with normotensive blood pressure who maintained normal blood pressure and did not have proteinuria were enrolled as controls. The women who were delivered preterm were delivered because of preterm labor, preterm rupture of membranes, incompetent cervix, or vaginal bleeding with possible abruptio placentae. The study group was classified into patients with gestational hypertension (n = 5), preeclampsia (n = 18), severe preeclampsia (n = 14), and HELLP syndrome (n = 7). To determine if the sample size had sufficient power, calculations were performed to give 80% power at the 5% significance level based on the validation study [15]. The required effect size was calculated for a minimum sample size of n = 51, which is smaller than the final 58 women analyzed.

### 2.2. Clinical Diagnosis of Pregnancy-Related Hypertensive Disorders

According to the American College of Obstetricians and Gynecologists (ACOG) diagnostic criteria [16], gestational hypertension is defined as a blood pressure ≥ 140/90 mmHg without proteinuria or with proteinuria of no greater than trace levels after 20 weeks of gestation. Preeclampsia was defined as a blood pressure ≥ 140/90 mmHg with proteinuria of 1+ on dipstick urinalysis for two samples taken 6 h apart or >0.3 g in a 24 h urine collection. The pregnant women with preeclampsia had at least one or more risk factors, such as headache, visual or cerebral disturbance, elevated liver enzymes, thrombocytopenia, dyspnea due to pulmonary edema, progressive renal failure, asystolic blood pressure ≥ 160 mmHg and/or a diastolic blood pressure ≥ 110 mmHg, and classified as having severe PE. According to the Tennessee classification [17], HELLP syndrome was defined as an increase in serum lactate dehydrogenase (LDH) of more than 600 IU/L, an increase in transaminase enzymes of 70 IU/L or more, and a platelet count of less than 100,000/mm^3^.

Patients with a history of preexisting medical disorders, such as type II diabetes mellitus, chronic hypertension, autoimmune disease, renal disease, and liver disease, as well as any cardiovascular, thyroid, or other endocrinological disorder were excluded from the study. Women with multiple pregnancies, fetal malformations, fetal chromosomal abnormalities, and unavailable or incomplete medical records were also excluded.

### 2.3. Clinical and Laboratory Data

Demographic information and clinical and laboratory data were collected at the time of hospitalization for delivery; among the variables, only the pregnancy-associated plasma protein-A (PAPP-A) data were collected from 11 weeks to 13 + 6 weeks of gestation. The other blood and urine results were obtained from routine blood and urine tests performed during hospitalization for birth indications.

The data were obtained from the patient’s electronic medical records and were ascertained by 2 independent researchers. In this study, the GFR (glomerular filtration rate) was estimated using the Modification of Diet in Renal Disease (MDRD: GFR = 175 (serum creatinine [µmol/l] × 0.011312)^−1.154^ × age^−0.205^ × 0.742) [18].

### 2.4. Histopathological Examination of the Placenta

Placental examination was performed according to the recommendations of the American Society of Clinical Pathologists [19]. We investigated syncytial knots, villous agglutination, fibrin deposition, distal villous hyperplasia, muscularized basal plate arteries, trophoblastic giant cells, placental septa, cell islands, placental infarct, inflammation (necrotizing chorioamnionitis), and thin umbilical cord. All histological samples were reviewed by a pathologist blinded to the clinical data and severity of disease except for the gestational age (which was needed to assess villous maturation). The sample preparation was performed based on standardized methods [20]. After the membranes and umbilical cord were removed, the placentas were weighed, measured, and sliced. Representative sections of the umbilical cord, membranes, placental parenchyma, and any abnormalities seen on gross specimens were submitted for standard histological examination. Formalin-fixed, paraffin-embedded tissues were processed and stained with hematoxylin and eosin (H&E) using routine staining. Every specimen that was reviewed had at least two sections taken from each of the cord, membranes, and placental parenchyma. The criteria for the investigated placental lesions were as follows:Increased syncytial knots: defined as increased syncytiotrophoblast nuclei along the stem villi or distal villi and knots in >20% of the villi at preterm (before 34 weeks) or 30% of the villi at term (after 38 weeks).Villous agglutination: defined as the clustering of villi with trophoblastic cohesion and bridging syncytial knots.Increased intervillous fibrin: defined as the focal or diffuse deposition of abnormal amounts of fibrin in intervillous or perivillous spaces.Distal villous hypoplasia: defined as small elongated distal villi of a single capillary loop with syncytiotrophoblasts. Focal with <30% on one slide, diffuse with >30% on >1 slide.Acute atherosis: defined as subintimal foamy macrophages in decidual vessels with fibrinoid necrosis.Muscularized basal plate arteries: defined as nontransformed basal plate arteries with a relatively small caliber.Trophoblastic giant cells: defined as multinucleated trophoblast giant cells in the decidua basalis.Placental septa: defined as two times the thickness of the adjacent maternal extravillous trophoblast layer.Cell islands: defined as aggregates of more than 50 extravillous trophoblasts. Increased cell islands were defined as more than 5 cell islands [21].Placental infarct: defined as villous ischemic necrosis from maternal vascular compromise.Inflammation (necrotizing chorioamnionitis): defined as abundant neutrophils in the amnion associated with neutrophil karyorrhexis.Thin umbilical cord: defined as a cord less than 8.0 mm of the maximum cross-sectional diameter.

### 2.5. Statistical Analysis

The following steps were taken:

(1) The women were subdivided into five groups depending on the severity of the pregnancy-related hypertensive disorder: normal group; gestational hypertension group; mild preeclampsia group; severe preeclampsia group; and HELLP syndrome group. Clinical and placental phenotypes were compared among groups using ANOVA when the data were normally distributed and the nonparametric Kruskal–Wallis test when the data were not normally distributed.

(2) A univariate linear analysis was performed to determine whether each variable was related to the severity of pregnancy-related hypertensive disorders by dividing it into clinical, laboratory, and histopathological factors.

(3) Multiple linear regression analyses were conducted to determine independent variables of the severity of pregnancy-related hypertensive disorders. The variables identified as associated in the univariate analysis at a level of less than 0.2 were included. Stepwise backward elimination and all subset regressions were performed to select the final model.

Statistical significance was defined as a *p*-value < 0.05. Statistical analyses were performed using R 4.0.3.

### 2.6. Ethics

The study protocol and the waiver of informed consent were approved by the Institutional Review Board (IRB) of Gyeongsang National University Changwon Hospital (serial number: GNUCH 22-08-015). All methods were performed in accordance with the relevant guidelines and regulations of the institution.

## 3. Results

### 3.1. Baseline Characteristics

We analyzed 58 women who were stratified into five groups according to their maternal hypertensive status and clinical severity, and their clinical, laboratory, and pathological characteristics are summarized in Table 1. Maternal age, body mass index (BMI), parity, and gestational age did not differ significantly among the groups. However, in the normotensive pregnancy group, the proportion of pregnancies conceived by assisted reproductive technology (ART) was significantly lower. Most of the laboratory factors were significantly different among the groups. Calcium levels, albumin levels, and estimated glomerular filtration rates were high in the normotensive group, and it was observed that the values decreased as the severity increased. On the other hand, uric acid levels, creatinine levels, and the urine protein/creatinine ratio (UPCR) were lower in the normotensive group, and significant differences were observed as the number of groups increased. PAPP-A levels were the highest in the normotensive group and the lowest in the HELLP group but were lower in the gestational hypertension and mild preeclampsia groups than in the severe preeclampsia group. There were no significant differences in the pathological factors among the five groups. 

### 3.2. Clinical and Laboratory Factor Model Predicting the Severity of Pregnancy-Related Hypertensive Disorders

We performed univariate and multivariate linear regression analyses with clinical and laboratory variables to predict the severity of pregnancy-related hypertensive disorders. When performing this analysis, blood pressure, blood concentrations of liver enzymes, serum creatinine concentrations and platelet counts, which are clinical criteria for severe preeclampsia [22], and levels of proteinuria were excluded. Univariate linear regression was used to examine the relationships among the variables except the above variables and the severity of gestational hypertension. The PAPP-A level in the late first trimester, serum uric acid level, serum albumin level, serum total bilirubin level, and the eGFR were significantly associated with the severity of pregnancy-related hypertensive disorders (Table 2). The prediction model constructed via multiple regression using the four most influential factors, namely, PAPP-A levels in late first trimester, serum albumin levels, serum total bilirubin levels, and estimated glomerular filtration rates, was finally selected as the model (Table 3, Figure 1). This model provided an R2 value of 0.373 and an adjusted R2 value of 0.3248.

### 3.3. Histopathological Factor Model Predicting the Severity of Pregnancy-Related Hypertensive Disorders

We performed univariate and multivariate linear regression analyses with histopathological variables to predict the severity of pregnancy-related hypertensive disorders (Table 4). In the multivariate analysis, it was found that among the histopathological factors, only increased syncytial knots had a significant correlation with the severity of pregnancy-related hypertensive disorders (Figure 2).

## 4. Discussion

In the present study, we assessed differences in clinical, laboratory, and placental histopathology variables through classification according to the severity of pregnancy-related hypertensive disorders.

To date, there have been very few studies on clinical and laboratory differences according to the severity of pregnancy-related hypertensive disorders, including gestational hypertension without proteinuria and the histopathological correlation of the placenta. We observed that the PAPP-A level measured in the first trimester, albumin levels, eGFR, total bilirubin levels measured near delivery, and placental syncytial knot status were associated with the severity of pregnancy-related hypertensive disorders in addition to blood pressure or liver or kidney function, among which albumin levels and increased placental syncytial knots showed a significant association.

### 4.1. Clinical and Laboratory Factors Related to the Severity of Pregnancy-Related Hypertensive Disorders

Albumin is a major protein component in plasma that maintains plasma oncotic pressure, which is lower in women with PE than in healthy pregnant women. Increased capillary permeability following endothelial damage is partly responsible for this finding. Based on this mechanism, several studies have been conducted with maternal serum albumin levels as predictors of the severity of PE. They reported that albumin levels less than 3 g/dL were commonly associated with the severity of preeclampsia and poor perinatal prognosis [10,23]. Additionally, in this study, the median serum albumin value significantly decreased as the severity of pregnancy-related hypertensive disorders increased.

PAPP-A is a metalloproteinase produced by the placental syncytial trophoblast and is a placental glycoprotein that positively regulates the metabolism of insulin-like growth factor (IGF) [24]. Consequently, it can affect the growth of the placenta and fetus. Previous studies on the association between PAPP-A values in early pregnancy and preeclampsia have been conducted, and relatively low PAPP-A values were observed in the preeclampsia group compared to the normal group [25,26]. These reports support the usefulness of PAPP-A levels as a biomarker for predicting the occurrence of PE. However, no consensus has been reached about the negative correlation between moderate severity and PAPP-A levels. While some results showed lower PAPP-A values in severe preeclampsia than mild preeclampsia [27,28], higher PAPP-A values were reported in severe preeclampsia that occurred after 32 weeks of gestation than in gestational hypertension and mild preeclampsia [29]. In this study, the PAPP-A values were significantly higher in the normotensive group, but there was no negative correlation with the increase in the severity of pregnancy-related hypertensive disorders. The PAPP-A value was the lowest in the HELLP patients but was lower in the gestational hypertension and mild preeclampsia patients than in the severe preeclampsia patients. This is presumed to be due to the inconsistency of gestational weeks and the inclusion of patients with poor pregnancy outcomes due to factors related to the placenta, such as placental abruption.

The GFR is the gold standard for renal functional assessment and is frequently used to determine the progression of kidney diseases. There have been a few reports showing that the GFR is associated with PE and small for gestational age [30,31]. In our study, there was a clear difference in the GFR between the normal group and the pregnancy-related hypertensive disorder group, and the results were observed to decrease as the severity increased. Because changes in renal function can affect both the mother and fetus, it is crucial to identify the risk of worsening renal status. Therefore, the authors think that it is important to evaluate the GFR as well as serum creatinine levels, which are currently used to evaluate the severity of PE.

To our knowledge, there are no studies related to the severity of serum bilirubin levels and the severity of pregnancy-related hypertensive disorders. However, there have been reports of an association with low serum bilirubin levels in PE and poor perinatal outcomes, which have also been seen in low-birth-weight infants [32,33]. In a Delphi survey [34], liver function tests were considered the third most important predictor of maternal and fetal complications in PE after blood pressure and proteinuria. In our study, serum bilirubin levels had the lowest median value in the normal group and the highest value in the HELLP syndrome group. More research on bilirubin is needed in the future.

### 4.2. Placental Histopathology Factors Related to the Severity of Pregnancy-Related Hypertensive Disorders

Pathophysiologically, PE is known to occur from (1) abnormal trophoblastic function or implantation, which leads to placental hypoxia, (2) immunologic maladaptation, and (3) genetic factors [35]. Although there is no specific diagnostic pathological feature for any form of hypertension in pregnancy, the most characteristic histopathological finding for PE is caused by MVM. MVM, in other words, maternal vascular underperfusion and uteroplacental insufficiency, is caused by abnormal maternal perfusion to the placenta. It consists of a group of findings, including small placenta by weight, distal villous hypoplasia, accelerated villous maturation, increased perivillous or intervillous fibrin, villous agglutination, and placental infarcts [35]. Among them, in this study, increased syncytial knots were significantly associated with the severity of pregnancy-related hypertensive disorders. Trophoblastic giant cells were observed in only two patients overall. Since superficial decidua could not always be included in the evaluation of each patient in the real-world placental gross examination, a larger study is needed. In our patients, an increase in syncytial knots was expressed as a percentage in the representative slide and was classified according to the gestational age [36]. Therefore, we think that it was the most suitable for quantitatively representing the features of MVM compared to other findings.

Regarding pregnancy-related hypertensive disorders, which are multisystem syndromes, we have not yet made great progress toward determining clinically useful predictions and have not achieved practical success in prevention or treatment. One possibility is that different subtypes of pregnancy-associated hypertensive disorders have different pathophysiological pathways [37]. Therefore, it is very important to classify pregnancy-related hypertensive disorders into several subtypes, rather than as a single disease, and to examine the clinical and histopathological differences. In that context, this study is meaningful.

A limitation of this study is that it was a single-center study with a small number of patients conducted for a relatively short period of time. Longer duration, larger, multicenter studies are needed to further confirm the role of the tested markers in the assessment of the severity of pregnancy-related hypertensive disorders and perinatal outcomes. Another limitation is that as an observational study, our results were susceptible to potential confounding factors. Although we controlled for potential confounders in multivariate analysis, there may be unidentified confounders that bias our results.

## 5. Conclusions

In conclusion, maternal albumin levels before delivery and placental syncytial knot status are closely related to disease severity in women with pregnancy-related hypertensive disorders. The combination of PAPP-A levels in the first trimester, albumin levels, eGFR, and total bilirubin levels just before delivery was optimal for predicting the severity of pregnancy-related hypertensive disorders. Thus, proper evaluation and interpretation of these markers will help guide clinical decisions and improve perinatal outcomes. This conclusion is based on a relatively small scale and observational study, and future large-scale prospective and experimental studies are needed to further consolidate the findings that these markers have clear value in assessing the risk and severity of pregnancy-related hypertensive disorders.

## Figures and Tables

**Figure 1 diagnostics-12-02188-f001:**
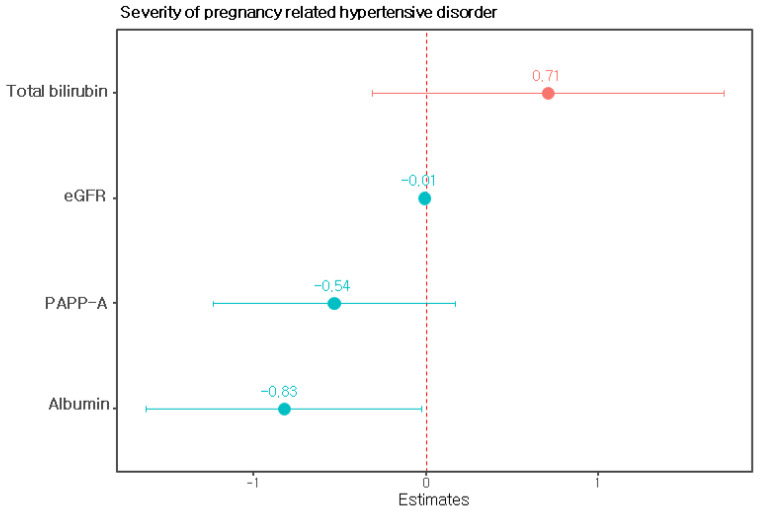
Regression coefficients of the model predicting the severity of pregnancy-related hypertensive disorders. Note: PAPP-A, pregnancy-associated plasma protein-A; eGFR, estimated glomerular filtration rate.

**Figure 2 diagnostics-12-02188-f002:**
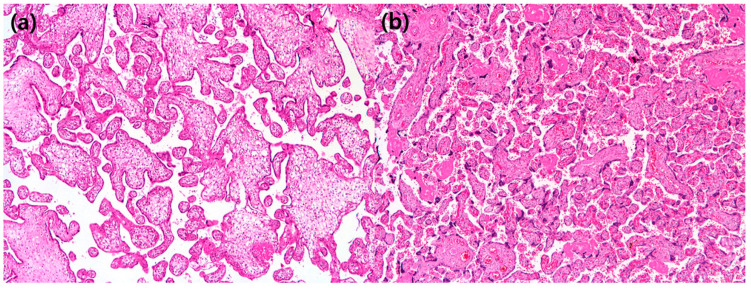
(**a**) In this patient with a preterm placenta, syncytial knots are barely seen (×40 hematoxylin and eosin); (**b**) in this patient with a preterm placenta, syncytial knots are rated as 90%, suggestive of “increased syncytial knots” (×40 hematoxylin and eosin).

**Table 1 diagnostics-12-02188-t001:** Frequencies of clinical and placental phenotypes.

	Group 1 Normal (n = 14)	Group 2 Gestational Hypertension (n = 5)	Group 3 Mild Preeclampsia (n = 18)	Group 4 Severe Preeclampsia (n = 14)	Group 5 HELLP Syndrome (n = 7)	*p*
Clinical variables						
Age, years	33.9 ± 4.7	34.0 ± 8.2	34.6 ± 3.9	34.5 ± 4.2	35.6 ± 3.2	0.437
Gestational age at delivery (weeks)	31.7 ± 4.5	36.9 ± 1.3	35.8 ± 1.3	33.2 ± 3.4	29.7 ± 2.1	0.507
Nulliparous, n (%)	7 (50.0%)	4 (80.0%)	13 (72.2%)	10 (71.4%)	4 (57.1%)	0.601
Pregnancy conceived by ART, n (%)	**4 (28.6%)**	**2 (40.0%)**	**12 (66.7%)**	**2 (14.3%)**	**3 (42.9%)**	**0.04**
Maternal height (cm)	161.8 ± 6.0	161.6 ± 3.2	163.6 ± 5.8	162.4 ± 5.4	162.3 ± 3.4	0.734
Maternal prepregnancy weight (kg)	61.0 ± 8.2	77.6 ± 17.8	70.6 ± 17.3	72.9 ± 16.8	58.8 ± 11.3	0.58
Maternal prepregnancy BMI (kg/m^2^)	23.4 ± 3.7	29.9 ± 7.7	24.8 ± 8.1	27.4 ± 5.0	22.4 ± 4.3	0.766
Maternal weight at birth (kg)	67.8 ± 6.7	90.2 ± 16.6	86.3 ± 18.2	84.3 ± 16.0	66.8 ± 11.6	0.349
Maternal BMI at birth (kg/m^2^)	26.0 ± 3.1	34.7 ± 7.3	32.1 ± 5.9	31.8 ± 4.5	25.4 ± 4.7	0.406
Systolic blood pressure (mmHg)	**137.9 ± 10.3**	**161.2 ± 4.9**	**174.2 ± 13.4**	**171.4 ± 17.5**	**159.4 ± 25.6**	**<0.001**
Diastolic blood pressure (mmHg)	**86.6 ± 8.2**	**97.4 ± 11.7**	**104.6 ± 7.9**	**110.3 ± 10.9**	**100.4 ± 15.4**	**<0.001**
Cerebral or visual symptoms, n (%)	0 (0.0%)	0 (0.0%)	1 (5.6%)	3 (21.4%)	2 (28.6%)	0.136
PAPP-A (MoM)	**1.7 ± 0.6**	**1.0 ± 0.2**	**1.0 ± 0.3**	**1.2 ± 0.4**	**0.9 ± 0.4**	**0.001**
Urine protein to creatinine ratio	**0.2 ± 0.1**	**0.3 ± 0.1**	**1.2 ± 1.2**	**3.3 ± 3.6**	**3.9 ± 4.6**	**<0.001**
Hemoglobin (g/dL)	11.3 ± 1.7	12.7 ± 1.2	11.4 ± 1.7	12.6 ± 1.6	12.1 ± 2.4	0.169
Platelets (× 10^9^/L)	220.1 ± 48.6	180.8 ± 57.5	200.8 ± 94.0	209.3 ± 63.3	142.4 ± 60.3	0.119
Segmented neutrophils (%)	74.1 ± 9.0	72.0 ± 7.3	72.1 ± 9.4	71.7 ± 8.1	72.0 ± 7.0	0.466
Lymphocytes (%)	18.9 ± 7.5	20.4 ± 6.1	20.1 ± 8.4	19.9 ± 7.2	20.7 ± 5.7	0.629
Calcium (mg/dL)	**8.9 ± 0.4**	**8.8 ± 0.4**	**8.7 ± 0.4**	**8.8 ± 0.7**	**8.3 ± 0.7**	**0.036**
Uric acid (mg/dL)	**4.5 ± 1.2**	**5.1 ± 2.3**	**6.5 ± 1.5**	**6.6 ± 1.4**	**5.5 ± 2.2**	**0.01**
Protein (g/dL)	6.3 ± 0.8	5.7 ± 1.1	5.7 ± 0.5	6.0 ± 0.6	5.7 ± 1.1	0.203
Albumin (g/dL)	**3.4 ± 0.4**	**3.2 ± 0.5**	**3.0 ± 0.3**	**3.1 ± 0.3**	**2.8 ± 0.5**	**0.002**
Total bilirubin (mg/dL)	**0.4 ± 0.1**	**0.5 ± 0.1**	**0.6 ± 0.4**	**0.5 ± 0.3**	**0.8 ± 0.4**	**0.008**
AST (U/L)	**23.9 ± 20.6**	**21.0 ± 5.0**	**28.8 ± 12.8**	**32.7 ± 34.5**	**126.1 ± 129.3**	**0.003**
ALT (U/L)	**23.1 ± 40.8**	**13.4 ± 3.8**	**20.5 ± 16.0**	**26.4 ± 32.7**	**155.3 ± 173.0**	**0.008**
Creatinine (mg/dL)	**0.5 ± 0.1**	**0.5 ± 0.1**	**0.6 ± 0.1**	**0.6 ± 0.1**	**0.7 ± 0.3**	**0.01**
gamma-GT (U/L)	**14.7 ± 15.1**	**12.2 ± 2.0**	**13.3 ± 6.2**	**14.3 ± 10.5**	**78.9 ± 66.6**	**0.003**
Potassium (mmol/L)	**3.9 ± 0.3**	**4.1 ± 0.2**	**4.0 ± 0.5**	**4.1 ± 0.4**	**3.0 ± 1.3**	**0.048**
Magnesium (mg/dL)	**1.8 ± 0.2**	**1.8 ± 0.1**	**1.9 ± 0.2**	**2.2 ± 0.8**	**2.1 ± 0.2**	**0.037**
eGFR (mL/min/1.73 m^2^)	**159.8 ± 27.2**	**145.8 ± 26.8**	**118.4 ± 29.3**	**116.2 ± 25.8**	**124.1 ± 57.9**	**0.001**
Histopathological variables						
Increased syncytial knots	10 (71.4%)	4 (80.0%)	17 (94.4%)	13 (92.9%)	7 (100.0%)	0.208
Villous agglutination	4 (28.6%)	4 (80.0%)	8 (44.4%)	5 (35.7%)	1 (14.3%)	0.178
Increased intervillous fibrin	0 (0.0%)	1 (20.0%)	4 (22.2%)	1 (7.1%)	1 (14.3%)	0.365
Distal villous hypoplasia						0.652
Focal	8 (57.1%)	2 (40.0%)	14 (77.8%)	8 (57.1%)	4 (57.1%)	
Diffuse	6 (42.9%)	3 (60.0%)	3 (16.7%)	5 (35.7%)	3 (42.9%)	
Acute atherosis	1 (7.1%)	1 (20.0%)	0 (0.0%)	2 (14.3%)	0 (0.0%)	0.357
Muscularized basal plate arteries	1 (7.1%)	1 (20.0%)	0 (0.0%)	2 (14.3%)	0 (0.0%)	0.357
Trophoblastic giant cells	1 (7.1%)	1 (20.0%)	0 (0.0%)	0 (0.0%)	0 (0.0%)	0.193
Placental septa	10 (71.4%)	3 (60.0%)	16 (88.9%)	12 (85.7%)	5 (71.4%)	0.511
Cell islands	4 (28.6%)	1 (20.0%)	8 (44.4%)	8 (57.1%)	4 (57.1%)	0.407
Placental infarct	6 (42.9%)	1 (20.0%)	5 (27.8%)	6 (42.9%)	3 (42.9%)	0.774
Inflammation (necrotizing chorioamnionitis)	1 (7.1%)	0 (0.0%)	1 (5.6%)	0 (0.0%)	0 (0.0%)	0.783
Thin umbilical cord	0 (0.0%)	0 (0.0%)	0 (0.0%)	0 (0.0%)	0 (0.0%)	

Note: Values are presented as the means ± standard deviations for continuous variables and n (%) for categorical variables. ART, assisted reproductive technology; BMI, body mass index; PAPP-A, pregnancy-associated plasma protein-A; MoM, multiple of median; AST, aspartate aminotransferase; ALT, alanine aminotransferase; eGFR, estimated glomerular filtration rate; HELLP, hemolysis, elevated liver enzymes, and low platelet count; n, number of participants; %, percentage. The bold values indicate significant differences (*p* < 0.05).

**Table 2 diagnostics-12-02188-t002:** Univariable linear regression models for the clinical and laboratory variable selection.

Variable	Beta Coefficient	SE	*p*
Maternal prepregnancy BMI	0.03	0.13	0.835
Maternal BMI at birth	0.1	0.13	0.454
PAPP-A	**−1.21**	**0.32**	**<0.001**
Uric acid	0.33	0.13	**0.013**
Protein	−0.29	0.23	0.215
Albumin	**−1.29**	**0.39**	**0.002**
Total bilirubin	**1.53**	**0.53**	**0.006**
eGFR	**−0.43**	**0.12**	**<0.001**

Note: BMI, body mass index; PAPP-A, pregnancy-associated plasma protein-A; eGFR, estimated glomerular filtration rate; SE, standard error. The bold values indicate significant differences (*p* < 0.05).

**Table 3 diagnostics-12-02188-t003:** Multivariable linear regression model predicting the severity of pregnancy-related hypertensive disorders.

	Beta Coefficient	SE	*p*
(Intercept)	6.02	1.41	<0.001
PAPP-A	−0.54	0.35	0.132
Albumin	**−0.83**	**0.4**	**0.043**
Total bilirubin	0.71	0.51	0.171
eGFR	**−0.01**	**0**	**0.035**

R2 = 0.373, adjR2 = 0.3248, F = 7.74, *p* < 0.001, AIC = 179.96. The predicted value of severity of pregnancy-related hypertensive disorders was calculated by the equation: 6.02 − 0.54 × PAPP-A − 0.83 × albumin + 0.71 × total bilirubin − 0.01 × eGFR. Note: PAPP-A, pregnancy-associated plasma protein-A; eGFR, estimated glomerular filtration rate; SE, standard error. The bold values indicate significant differences (*p* < 0.05).

**Table 4 diagnostics-12-02188-t004:** Histopathological factors related to the severity of pregnancy-related hypertensive disorders by linear regression analysis.

	Univariate Analysis			Multivariate Analysis		
	Beta Coefficient	SE	*p*	Beta Coefficient	SE	*p*
Increased syncytial knots	**0.82**	**0.33**	**0.017**	**0.71**	**0.31**	**0.026**
Villous agglutination	0.04	0.23	0.872			
Increased intervillous fibrin	0.48	0.34	0.164	0.37	0.33	0.28
Acute atherosis	−0.19	0.45	0.667			
Trophoblastic giant cells	−0.96	0.61	0.12	−0.65	0.61	0.289
Placental infarct	−0.08	0.24	0.742			

Note: SE; standard error. The bold values are values with significant differences (*p* < 0.05).

## Data Availability

Not applicable.

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
