# Peer review of "Factors Associated with the Severity of Pregnancy-Related Hypertensive Disorder: Significance of Clinical, Laboratory, and Histopathological Features"

_diagnostics, 2022, doi:10.3390/diagnostics12092188_

Round 1
Reviewer 1 Report
The authors identified Total Bilirubin, eGFR, PAPP-A, Albumin and syncytial knots as markers that correlate with severity of pregnancy related hypertensive disorder. The overall study is interesting and agree with authors that further larger scale study with multiple medical centers will be needed to advance the work further. I only have a few comments / suggestions:
- Total Bilirubin was misspelled in Figure 1
- Not sure what Kalium is in Table 1 (is it something measured routinely at your medical center?)
- being a non-statistian, I am a bit confused about the way the data are presented. In the various linear (univariable vs multivariable) regression analysis, the statistical values are not presented the same way (Table 2-4). Beta Coefficient was presented in table 2 and 4, but not 3. T value was in table 3 but not 2 and 4. It may be easier for the readers if the data can be presented in a similar way if possible.
- In Table 3, it stated that the predicted value of severity was calculated by the equation "6.02-0.54*PAPP-A - 0.83*albumin + 0.71*total bilirubin - 0.01*eGFR". How did that equation gets generated? Is the goal to determine if the marker has stronger predictive value on its own (univariable regression analysis) vs together (multivariable regression analysis).
- Generally, I recommend further description of the statistical analysis, and elaborate on what the meaning of the statistical finding is to clinical practice. While it may be clear to the authors, most clinicians have not used advanced statistics for a long time. Therefore, most likely will not be able to fully appreciate the analysis to make it applicable to their practice.
Author Response
We appreciate the effort you have dedicated to providing insightful feedback on ways to strengthen our paper. Our answer based on your suggestion is provided attachment.
Please see the attachment.
Authors’ Reply to the Review Report
The authors identified Total Bilirubin, eGFR, PAPP-A, Albumin and syncytial knots as markers that correlate with severity of pregnancy related hypertensive disorder. The overall study is interesting and agree with authors that further larger scale study with multiple medical centers will be needed to advance the work further. I only have a few comments / suggestions:
- First, we appreciate the effort you have dedicated to providing insightful feedback on ways to strengthen our paper. Our answer based on your suggestion is provided below.
- Total Bilirubin was misspelled in Figure 1
- We apologize for the mistake. We corrected the spelling in Figure 1.
- Not sure what Kalium is in Table 1 (is it something measured routinely at your medical center?)
- We apologize for any confusion. We changed Kalium to Potassium in Table 1. Potassium is measured in routine clinical settings.
- being a non-statistian, I am a bit confused about the way the data are presented. In the various linear (univariable vs multivariable) regression analysis, the statistical values are not presented the same way (Table 2-4). Beta Coefficient was presented in table 2 and 4, but not 3. T value was in table 3 but not 2 and 4. It may be easier for the readers if the data can be presented in a similar way if possible.
- We appreciate your insightful comments. We apologize for any confusion. We fully agree with your opinion. We have revised Tables 2-4 in a similar format.
- In Table 3, it stated that the predicted value of severity was calculated by the equation "6.02-0.54*PAPP-A - 0.83*albumin + 0.71*total bilirubin - 0.01*eGFR". How did that equation gets generated? Is the goal to determine if the marker has stronger predictive value on its own (univariable regression analysis) vs together (multivariable regression analysis).
- Thank you for providing these insights. This equation was confirmed by multivariable regression analysis. It was not the goal of this study to determine whether together (multivariate regression analysis) had stronger predictive values.
- Generally, I recommend further description of the statistical analysis, and elaborate on what the meaning of the statistical finding is to clinical practice. While it may be clear to the authors, most clinicians have not used advanced statistics for a long time. Therefore, most likely will not be able to fully appreciate the analysis to make it applicable to their practice.
- We appreciate your insightful comments. You raised an important issue, and we fully agree with you. As per your comments, the statistical analysis has been described in more detail (page4, line160-174).
- Revision
The following steps were taken:
1) The women were subdivided into five groups depending on the severity of the pregnancy-related hypertensive disorder: normal group; gestational hypertension group; mild preeclampsia group; severe preeclampsia group; and HELLP syndrome group. Clinical and placental phenotypes were compared among groups using ANOVA when the data were normally distributed and the nonparametric Kruskal‒Wallis test when the data were not normally distributed.
2) A univariate linear analysis was performed to determine whether each variable was related to the severity of pregnancy-related hypertensive disorders by dividing it into clinical, laboratory and histopathological factors.
3) Multiple linear regression analyses were conducted to determine independent variables of the severity of pregnancy-related hypertensive disorders. The variables identified as associated in the univariate analysis at a level of less than 0.2 were included. Stepwise backward elimination and all subset regressions were performed to select the final model.
Reviewer 2 Report
This is a retrospective, one-center study, where authors tried to evaluate the association of maternal serum marker levels and placental histopathological changes with disease severity in pregnancy-related hypertensive disorders. They found that
maternal albumin levels before delivery and increased placental syncytial knots are important parameters closely related to the disease severity in women with pregnancy-related hypertensive disorder and that the combination of albumin, PAPP-A, total bilirubin, and eGFR levels appears to be optimal for predicting the pregnancy-related hypertensive disorder severity.
There are various points to be revisited, in order to improve the quality of this paper.
Among them:
1. The language and grammar need remodeling in the whole text.
2. The abstract section needs mild remodeling of the whole structure in most of the sections, in terms of direct synchronization.
3. The introduction section needs to contain the rationale of the paper, through the gap of the literature. For example, authors should indicate if there are data on placental histopathology and clinical and laboratory variables through classification according to the severity of pregnancy related hypertensive disorder, so that their paper appears meaningful.
4. Sample size calculation is missing.
5. Authors refer to a model; they should describe it more accurately, giving the rationale for example of the parameters included.
6. In the methods section, I wonder why authors did not choose a higher number of participants in the control group. Also, regression / subgroup analyses with regards to age, for example or the way of conception could improve the robustness of the results.
7. The discussion section should obtain a more structured format.
8. The limitations section should be enriched.
9. The final conclusion should be based on the exact findings of the study and its quality.
Author Response
We appreciate the effort you have dedicated to providing insightful feedback on ways to strengthen our paper. Our answer based on your suggestion is provided attachment.
Please see the attachment.
Authors’ Reply to the Review Report
This is a retrospective, one-center study, where authors tried to evaluate the association of maternal serum marker levels and placental histopathological changes with disease severity in pregnancy-related hypertensive disorders. They found that
maternal albumin levels before delivery and increased placental syncytial knots are important parameters closely related to the disease severity in women with pregnancy-related hypertensive disorder and that the combination of albumin, PAPP-A, total bilirubin, and eGFR levels appears to be optimal for predicting the pregnancy-related hypertensive disorder severity.
There are various points to be revisited, in order to improve the quality of this paper.
Among them:
- The language and grammar need remodeling in the whole text
- First, we appreciate the effort you have dedicated to providing insightful feedback on ways to strengthen our paper.
We apologize for any inconvenience in reading the paper caused by language issues and incorrect grammar. We received English proofreading when writing the paper, but we discussed your points with the proofreading team again and revised it once more.
- The abstract section needs mild remodeling of the whole structure in most of the sections, in terms of direct synchronization.
- We appreciate your insightful comments. As per your suggestion, the abstract has been modified for synchronization.
- Revision: To evaluate the association of maternal clinical and laboratory features and placental histopathological changes with disease severity in pregnancy-related hypertensive disorders. From January 2021 to December 2021, clinical, and laboratory data at the time of delivery and histopathological features of the placenta were collected from pregnant women with pregnancy-related hypertensive disorders at a single institution. The women were classified according to the pregnancy-related hypertensive disorder clinical severity, and each variable was compared accordingly. Gestational age-matched normotensive groups were also compared. Univariate and multivariate regression analyses were used to identify factors influencing pregnancy-related hypertensive disorder severity. Fifty-eight pregnancies were analyzed. Maternal albumin levels before delivery (beta coefficient -0.83, p = 0.043) and increased placental syncytial knots (beta coefficient 0.71, p = 0.026) are important parameters that are closely related to disease severity in women with pregnancy-related hypertensive disorders. The combination of albumin, PAPP-A, total bilirubin, and eGFR levels appears to be optimal for predicting pregnancy-related hypertensive disorder severity.
- The introduction section needs to contain the rationale of the paper, through the gap of the literature. For example, authors should indicate if there are data on placental histopathology and clinical and laboratory variables through classification according to the severity of pregnancy related hypertensive disorder, so that their paper appears meaningful.
- We appreciate your insightful comments. As per your suggestion, we have further described the existing studies related to placental pathology and the severity of gestational hypertensive disease (page2, line50-63).
- Revision:
Because the decision to terminate a pregnancy depends on the severity of the findings, previous studies have investigated biomarkers such as liver or kidney function, ultrasound results or clinical parameters for the severity of the disease [6-10] and reported that these markers are affected by the onset and severity of PE. However, indications for immediate or delayed delivery depend largely on the clinician's experience. Therefore, the more informative the markers for the criteria in judging PE severity, the more supportive the criteria can be in determining the time of delivery in clinical practice.
The association between PE and placental histopathological lesions, mostly represented by maternal vascular malperfusion (MVM) lesions, has been studied [11]. Only a few studies have investigated the association between the severity of PE and the findings of placental histopathology, even classifying it into two types, mild and severe PE [12] or early-onset and late-onset PE [13], and none of these studies looked for a correlation with gestational hypertension without proteinuria and HELLP syndrome.
- Roberts, J.M.; Escudero, C. The placenta in preeclampsia. Pregnancy Hypertension: An International Journal of Women's Cardiovascular Health 2012, 2, 72-83.
- VINNARS, M.T.; Nasiell, J.; Ghazi, S.; Westgren, M.; Papadogiannakis, N. The severity of clinical manifestations in preeclampsia correlates with the amount of placental infarction. Acta obstetricia et gynecologica Scandinavica 2011, 90, 19-25.
- Moldenhauer, J.S.; Stanek, J.; Warshak, C.; Khoury, J.; Sibai, B. The frequency and severity of placental findings in women with preeclampsia are gestational age dependent. American journal of obstetrics and gynecology 2003, 189, 1173-1177.
- Sample size calculation is missing.
Thank you for providing these insights. However, the study was a retrospective study, and we thought that the sample size calculation was not essential, and we tried to secure as many research subjects as possible.
- Authors refer to a model; they should describe it more accurately, giving the rationale for example of the parameters included.
Thank you for providing these insights. We have revised the text to reflect your suggestion (page8, line213-226).
- We performed univariate and multivariate linear regression analyses with clinical and laboratory variables to predict the severity of pregnancy-related hypertensive disorders. When performing this analysis, blood pressure, blood concentrations of liver enzymes, serum creatinine concentrations and platelet counts, which are clinical criteria for severe preeclampsia[21], and levels of proteinuria were excluded. Univariate linear regression was used to examine the relationships among the variables except the above variables and the severity of gestational hypertension. The PAPP-A level in the late first trimester, serum uric acid level, serum albumin level, serum total bilirubin level and the eGFR were significantly associated with the severity of pregnancy-related hypertensive disorders (Table 2). The prediction model constructed via multiple regression using the four most influential factors, PAPP-A levels in late first trimester, serum albumin levels, serum total bilirubin levels and estimated glomerular filtration rates, was finally selected as the model (Table 3, Figure 1). This model provided an R2 value of 0.373 and an adjusted R2 value of 0.3248.
- Gestational Hypertension and Preeclampsia: ACOG Practice Bulletin, Number 222. Obstetrics & Gynecology 2020, 135, e237-e260; DOI:10.1097/aog.0000000000003891.
- In the methods section, I wonder why authors did not choose a higher number of participants in the control group. Also, regression / subgroup analyses with regards to age, for example or the way of conception could improve the robustness of the results.
In the methods section, I wonder why authors did not choose a higher number of participants in the control group. Also, regression / subgroup analyses with regards to age, for example or the way of conception could improve the robustness of the results.
- Thank you for providing these insights. Although it is a limitation of our study, this study was a retrospective study conducted at a single institution for one year. The institution that conducted the study did not routinely perform placental biopsy every time in a pregnancy without problems. Among the cases in which placental biopsy was performed, the maximum number of cases was enrolled in this study, matching gestational age with the hypertensive group.
- We completely agree with your opinion. In this study, the number of cases was small for subgroup analysis. This content was described in addition to the limitation (page11, line340-342).
- Revision: As an observational study, our results were susceptible to potential confounding. Although we controlled for potential confounders in multivariate analysis, there may be unidentified confounders that bias our results.
- The discussion section should obtain a more structured format.
We agree with your comments and have tried to improve the readability by subheading the discussion sections (page10, line264 , page11, line310 )..
- Clinical and laboratory factors related to the severity of pregnancy-related hypertensive disorders
- Placental histopathology factors related to the severity of pregnancy-related hypertensive disorders
- The limitations section should be enriched.
Thank you for your suggestion. We agree and added content to the limitations section (page11, line340-342)..
- Revision: As an observational study, our results were susceptible to potential confounding. Although we controlled for potential confounders in multivariate analysis, there may be unidentified confounders that bias our results.
- The final conclusion should be based on the exact findings of the study and its quality
Thank you for providing these insights. We agree with your comments and have revised the conclusions (page12, line344-351).
- Original : In conclusion, changes in PAPP-A levels in the first trimester, albumin, eGFR, total bilirubin levels near delivery and placental syncytial knot status may play a role in the pathogenesis of pregnancy-related hypertensive disorders or may function in the devel-opment of pregnancy-related hypertensive disorders. Our data suggest that these markers may have potential value in assessing the risk and severity of pregnancy-related hyper-tensive disorder. Future large-scale prospective studies should further clarify the role these parameters play in predicting the risk and severity of pregnancy-related hypertensive dis-orders.
- Revision: In conclusion, maternal albumin levels before delivery and placental syncytial knot status are closely related to disease severity in women with pregnancy-related hypertensive disorders. The combination of PAPP-A levels in the first trimester, albumin levels, eGFR, and total bilirubin levels just before delivery was optimal for predicting the severity of pregnancy-related hypertensive disorders. Future large-scale prospective studies as well as experimental studies are needed to further consolidate the results that these markers are of clear value in assessing the risk and severity of pregnancy-related hypertensive disorders.
Round 2
Reviewer 2 Report
I think that most suggestions have been addressed. With 2 exceptions:
1. ALL clinical studies should accompanied by sample size calculation a priori, in order to eliminate -as much as possible- the high risk of bias that are linked per se.
2. The final conclusion should be expressed through the limitations and especially through the quality of the study.
Author Response
We appreciate the time and effort you have dedicated to providing insightful feedback on ways to strengthen our paper.
We have enclosed a list of the changes, presented as point-by-point responses to the comments.
